# CRPA: Curriculum-driven Reinforcement Pre-Alignment for Domain-Adaptive Vision-Language Models

## Abstract

Vision-Language Models (VLMs) demonstrate remarkable general-purpose capabilities but often fall short in specialized domains such as medical imaging or geometric problem-solving. Supervised Fine-Tuning (SFT) can enhance performance within a target domain, but it typically causes catastrophic forgetting, limiting its utility as a general AI agent. The central challenge, therefore, is to adapt VLMs to new domains while preserving their general-purpose capabilities. Continual pretraining is effective for expanding knowledge in Large Language Models (LLMs), but it is less feasible for VLMs due to prohibitive computational costs and the unavailability of pretraining data for most open-source models. This necessitates efficient post-training adaptation methods. Reinforcement learning (RL)–based approaches such as Group Relative Policy Optimization (GRPO) have shown promise in preserving general abilities, yet they often fail in domain adaptation scenarios where the model initially lacks sufficient domain knowledge, leading to optimization collapse. To bridge this gap, we propose **C**urriculum-driven **R**einforcement **P**re-**A**lignment (**CRPA**), a novel post-training paradigm that introduces a curriculum-aware progressive modulation mechanism. In the early phase, CRPA applies partial output constraints to safely expose the model to new domain concepts. As the model's domain familiarity increases, training gradually transitions to full generation optimization, refining responses and aligning them with domain-specific preferences. This staged adaptation balances domain knowledge acquisition with the preservation of general multimodal capabilities. Extensive experiments across specialized domains (e.g., OpenI for medical imaging and Geo170K for geometry) and general benchmarks (e.g., COCO Captions) validate the effectiveness of CRPA. Results show that CRPA achieves domain-specific performance competitive with SFT while significantly outperforming SFT in retaining general multimodal understanding, establishing a practical pathway toward building high-performing, domain-adaptive VLMs.

## 1 Introduction

Vision-Language Models (VLMs) (Bai et al., 2025; Wang et al., 2024a; Bai et al., 2023; Wang et al., 2025b; Zhu et al., 2025; Chen et al., 2024b; Wang et al., 2024b; Gao et al., 2024; Liu et al., 2024; 2023a) have achieved remarkable progress in unifying visual perception with natural language understanding, enabling powerful capabilities across tasks such as image captioning, visual question answering (VQA), and multimodal dialogue (Bai et al., 2025; Wang et al., 2024a; Bai et al., 2023). Large-scale pre-training on web-scale multimodal corpora equips these models with broad general-purpose competencies, making them versatile foundation models. However, real-world deployment often requires domain adaptation, where VLMs must acquire specialized knowledge (e.g., medical, scientific, or industrial contexts) while retaining their general reasoning abilities. This dual requirement—learning new domain knowledge without forgetting existing skills—poses a central challenge for post-training adaptation. Current post-training adaptation strategies (Gekhman et al., 2024; Guo et al., 2023; Hu et al., 2021; Lambert, 2025; Rafailov et al., 2023) can be broadly divided into two categories. Supervised Fine-Tuning (SFT) directly injects domain knowledge through imitation learning from data of target domain. While effective in transferring specialized expertise, SFT

is prone to catastrophic forgetting, as domain-specific signals overwrite previously acquired general capabilities (Shenfeld et al., 2025). Although numerous works have focused on mitigating this issue—including incremental learning methods (Li & Hoiem, 2016; Kirkpatrick et al., 2016; Rebuffi et al., 2016), as well as Test-Time Adaptation (TTA) approaches (MA et al., 2023; Niu et al., 2023; Yang et al., 2024)—their core designs are often ill-suited for fine-tuning large multimodal models. These methods typically rely on one of three strategies: fine-tuning only a subset of model parameters (Niu et al., 2023), imposing regularization constraints to preserve pre-trained knowledge (Li & Hoiem, 2016), or replaying historical data from the source domain (Rebuffi et al., 2016). However, in the context of multimodal large model fine-tuning, accessing pre-trained data is generally infeasible due to privacy, storage, or licensing constraints. Moreover, regularization-based approaches merely achieve a suboptimal trade-off between learning domain-specific knowledge and retaining general capabilities, often failing to yield high performance in both objectives simultaneously. In contrast, preference-based Reinforcement Learning (RL) methods such as Group Relative Policy Optimization (GRPO) (Shao et al., 2024) preserve general capabilities through KL-regularized optimization. Yet, these methods implicitly assume that the pre-trained model already possesses non-trivial domain knowledge (Wang et al., 2025a). When this assumption fails—common in low-resource or highly specialized domains—the model struggles to generate meaningful outputs, leading to optimization collapse.

While SFT facilitates knowledge transfer and RL helps preserve generalization, their respective shortcomings reveal a fundamental limitation: neither SFT nor RL-based adaptation alone can simultaneously guarantee stability, effective domain transfer, and the preservation of general-purpose capabilities. To bridge this gap, we argue that domain adaptation should be progressive rather than monolithic. Specifically, the model should first be pre-aligned to establish a minimal yet stable grounding in the target domain, and then reinforcement-aligned to refine its behavior with richer preference signals.

In this paper, we propose **C**urriculum-**D**riven **R**einforcement **P**re-Alignment (**CRPA**), a novel post-training framework that unifies domain knowledge acquisition with preference alignment under a curriculum-driven design. CRPA introduces a progressive modulation mechanism that transitions smoothly from constrained imitation to full generation optimization, avoiding optimization collapse while mitigating forgetting. To further enhance adaptability, CRPA incorporates two curriculum-inspired modules: Curriculum Progress Perception (CPP) and Curriculum Difficulty Perception (CDP). CPP regulates answer-prefix injection and reward threshold scheduling to bootstrap stable signals in early training, while CDP dynamically prioritizes difficult samples to maximize learning benefits and prevent overfitting. Extensive experiments on domain-specific VQA benchmarks demonstrate that CRPA not only achieves superior domain alignment but also preserves broad general-purpose multimodal capabilities, outperforming existing SFT- and RL-based approaches.

## 2 RELATED WORKS

### 2.1 VISION LANGUAGE MODELS (VLMs)

Vision Language Models (VLMs) (Bai et al., 2025; Wang et al., 2024a; Bai et al., 2023; Wang et al., 2025b; Zhu et al., 2025; Chen et al., 2024b; Wang et al., 2024b; Gao et al., 2024; Liu et al., 2024; 2023a) have significantly advanced cross-modal intelligence by integrating text and image modalities, progressing through three key phases. In the foundational phase, early models like CLIP (Radford et al., 2021) and ViT-BERT (Li et al., 2021b) bridged the modal gap between text and images, enabling tasks like zero-shot transfer and visual grounding. Subsequent models such as ALBEF (Li et al., 2021a) and FLAVA (Singh et al., 2022) further refined alignment techniques for better semantic consistency. The second phase focused on enhancing general capabilities through instruction tuning, with models like LLaVA (Liu et al., 2023b) and Flan-V5 (Chung et al., 2022) improving cross-modal reasoning and task handling. Recent developments include InternVL (Wang et al., 2025b; Zhu et al., 2025; Wang et al., 2024b), an open-source Vision Language Models (VLMs) series, and the Qwen series (Bai et al., 2025; Wang et al., 2024a; Bai et al., 2023), both pushing the boundaries of multimodal understanding with advanced visual encoders and innovative techniques for handling high-resolution images, multimodal rotation, and tool usage. Despite these advances, VLMs continue to struggle with domain-specific adaptation. In specialized settings such as medical imaging or scientific problem-solving, they often fail to recognize domain-specific concepts or adapt to nuanced visual features. To this end, post-training methods have emerged as a promising

direction, offering lightweight yet effective mechanisms for adapting VLMs to specialized domains without retraining from scratch.

## 2.2 POST-TRAINING FOR VLMS

Post-training techniques, primarily SFT and RL, have been central to adapting VLMs to domain-specific tasks (Kumar et al., 2025; Chu et al., 2025; Lai et al., 2025; Li et al., 2025). SFT enables task-specific learning but often leads to catastrophic forgetting of the general knowledge learned during pre-training, especially when fine-tuning is performed on domain-specific data (Duan et al., 2024; Dong et al., 2025; Chen et al., 2024a). Parameter-efficient approaches such as QLoRA (Dettmers et al., 2023), LoRA (Hu et al., 2021), Adapters (Hu et al., 2023), and Prompt/Prefix Tuning (Lester et al., 2021; Li & Liang, 2021) alleviate computational burdens by updating only a subset of parameters, yet they remain prone to overfitting and limited transferability. In contrast, RL-based methods such as Group Relative Policy Optimization (GRPO) (Shao et al., 2024), Domain Adaptive Policy Optimization (DAPO) (Yu et al., 2025), and Group Sequence Policy Optimization (GSPO) (Zheng et al., 2025a) enhance adaptability by leveraging dynamic feedback and optimizing sequential decision-making. These methods are effective in preserving general capabilities by regularizing the model's output with reward signals, but they assume that the pre-trained model already possesses non-trivial domain knowledge. Traditional RL approaches such as Proximal Policy Optimization (PPO) (Schulman et al., 2017), which require training both the policy and critic models (Lambert, 2025), impose high computational costs. Recent alternatives like GRPO and Direct Preference Optimization (DPO) (Rafailov et al., 2023) reduce this burden by removing the need for a separate critic, thereby simplifying training. Nevertheless, existing RL paradigms still struggle to adapt efficiently when the model begins with limited domain expertise. The commonly adopted "SFT-then-RL" pipeline (Shao et al., 2024) partially alleviates this issue by stabilizing the reward signal in early training. However, recent findings from CHORD (Zhang et al., 2025) reveal a fundamental flaw: SFT disrupts the pretrained model's internal structures, causing temporary degradation of general capabilities, while subsequent RL fails to recover domain adaptation—often performing worse than direct RL. These limitations underscore the pressing need for more efficient and scalable post-training strategies that can jointly achieve domain specialization and general capability preservation.

## 3 PRELIMINARY

### 3.1 PROBLEM FORMULATION

Consider a pre-trained VLM denoted as $\pi_{\text{pre}}$, which exhibits strong general-purpose multimodal capabilities. We are given a target domain-specific dataset $\mathcal{D}_{\text{target}} = \{(\mathbf{x}_i, y_i)\}_i^N$, where $\mathbf{x}_i = (\text{image}_i, \text{prompt}_i)$ represents a multimodal input (image paired with a task prompt) and $y_i$ is the ground-truth response containing domain-specific knowledge. The objective is to adapt $\pi_{\text{pre}}$ into a domain-adapted model $\pi_\theta$ through a post-training procedure, such that $\pi_\theta$ achieves high performance on $\mathcal{D}_{\text{target}}$ while maximally preserving the general capabilities of $\pi_{\text{pre}}$. This defines the fundamental challenge in domain-adaptive VLM alignment: how to integrate novel domain knowledge without forgetting previously acquired knowledge.

### 3.2 LIMITATIONS OF EXISTING METHODS

**Supervised Fine-Tuning (SFT).** SFT adapts model parameters $\theta$ by maximizing the likelihood of expert demonstrations in $\mathcal{D}_{\text{target}} = \{(\mathbf{x}_i, y_i)\}_{i=1}^N$. Its objective is:

$$\mathcal{J}_{\text{SFT}}(\theta) = \mathbb{E}(\mathbf{x}, y) \sim \mathcal{D}_{\text{target}} \left[ \sum_{t=1}^{|y|} \log \pi_\theta(y_t \mid \mathbf{x}, y_{<t}) \right], \qquad (1)$$

where $y_{<t}$ denotes the prefix tokens of $y$. While SFT effectively injects domain-specific knowledge via imitation learning, it relies exclusively on supervised labels—though this reliance does not directly cause catastrophic forgetting. Instead, the core driver is the distributional gap between SFT and pretraining data. When target domain data (for SFT) differs substantially from pretraining data, SFT-induced retraining aligns the model with the target domain distribution, leading to misalignment with pretraining data and subsequent catastrophic forgetting. Previously acquired capabilities are overwritten by domain-specific information, undermining VLMs' generalization.

**Group Relative Policy Optimization (GRPO).** GRPO and other Preference-based RL methods attempt to align models with human preferences while mitigating forgetting through regularization. The GRPO incorporates a KL-divergence penalty to constrain the updated policy $\pi_\theta$ from deviating excessively from a reference policy $\pi_{\theta_{\text{ref}}}$ (typically the initial pre-trained model $\pi_{\theta_{\text{pre}}}$). Formally, given a input $x$ and a group of $G$ responses $O = \{o_1, o_2, \ldots, o_G\}$ sampled from the old policy $\pi_{\theta_{\text{old}}}$. The GRPO objective maximizes the expected clipped advantage for each token $o_{i,t}$ in response $o_i$:

$$\mathcal{J}_{\text{GRPO}}(\theta) = \mathbb{E}_{x \sim P(X), \, \{o_i\}_{i=1}^G \sim \pi_{\theta_{\text{old}}}(O|q)}$$

$$\frac{1}{G} \sum_{i=1}^{G} \frac{1}{|o_i|} \sum_{t=1}^{|o_i|} \left\{ \min\left( \rho_{i,t}(\theta)\hat{A}_{i,t}, \text{clip}\left(\rho_{i,t}(\theta), 1 - \varepsilon, 1 + \varepsilon\right)\hat{A}_{i,t} \right) - \gamma \mathbb{D}_{\text{KL}}\left[\pi_\theta \| \pi_{\text{ref}}\right] \right\}, \tag{2}$$

where the $\rho_{i,t}(\theta) = \frac{\pi_\theta(o_{i,t}|x, o_{i,<t})}{\pi_{\theta_{\text{old}}}(o_{i,t}|x, o_{i,<t})}$ is the importance sampling ratio, $\epsilon$ is the clip factor, $\gamma$ controls the strength of KL regularization, and $\hat{A}_{i,t} = \frac{r_i - \text{mean}(\mathbf{r})}{\text{std}(\mathbf{r})}$ is the standardized advantage for token $o_{i,t}$, computed from the group rewards $\mathbf{r} = \{r_1, r_2, \ldots, r_G\}$.

GRPO alleviates forgetting by combining preference-based optimization with KL regularization. However, it presupposes that the pretrained model already holds non-trivial knowledge of the target domain. If the initial model $\pi_{\text{pre}}$ possesses limited knowledge of the target domain, it cannot generate responses of sufficient quality to yield informative reward signals, which is also known as *optimization collapse*.

### 3.3 MOTIVATION FOR CRPA

Domain-adaptive alignment of VLMs entails a dual objective: injecting novel domain knowledge while preserving general-purpose capabilities. This poses a substantial challenge for existing post-training paradigms. On one hand, SFT is effective at incorporating domain knowledge but inevitably induces catastrophic forgetting of general skills. On the other hand, RL-based methods such as GRPO emphasize preserving broad competencies through regularization, yet often suffer from optimization collapse when the model lacks sufficient prior knowledge of the target domain. Consequently, neither purely supervised nor purely reinforcement-based adaptation can simultaneously ensure stability, effective domain transfer, and robust generalization. This limitation motivates a progressive adaptation strategy, wherein the model is first pre-aligned to safely acquire foundational domain concepts and subsequently reinforcement-aligned to refine its behavior with richer preference signals. Building on this perspective, we propose Curriculum-Driven Reinforcement Pre-Alignment (CRPA)—a paradigm that introduces a progressive modulation mechanism to dynamically coordinate the training objective, enabling a smooth transition from constrained imitation learning to full reward-driven optimization as the model's domain familiarity evolves.

## 4 METHOD

### 4.1 OVERVIEW OF CRPA

CRPA is a post-training paradigm that unifies domain knowledge acquisition with preference alignment under a curriculum-driven framework. As illustrated in Figure 1, CRPA builds upon GRPON (GRPO for Non-Deep-Thinking Models) as the RL backbone for VQA-style tasks. A key design is progressive modulation mechanism, which decomposes adaptation into two coordinated phases:

**Pre-Alignment Phase:** At the early stages, when the model lacks sufficient domain knowledge, CRPA employs partial output constraints to mitigate the challenges of sparse rewards and unavailable preference signals. By introducing domain concepts in a controlled manner (e.g., through partial answer exposure), this phase ensures valid response generation and bootstraps the model's initial domain competence.

**Reinforcement Alignment Phase:** Once the model has attained a foundational understanding of the target domain, the training process gradually shifts toward full reward-driven optimization. Here, constraints are relaxed, and the model refines its responses based on reinforcement signals, achieving stronger preference alignment and higher task performance.

To enable smooth transitions, CRPA incorporates two curriculum-inspired modules: Curriculum Progress Perception (CPP) and Curriculum Difficulty Perception (CDP). We clarify the fundamen-

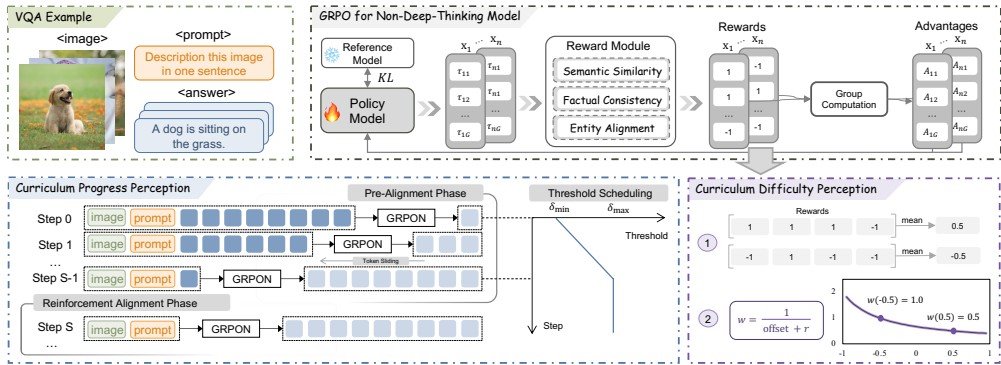

Figure 1: The Overview of CRPA. CRPA is a post-training framework that integrates domain knowledge acquisition with preference alignment in a curriculum-driven manner, built upon the GRPON (GRPO for Non-Deep-Thinking Models) framework for VQA-style tasks. It consists of two phases: Pre-Alignment, which introduces domain concepts with controlled constraints to bootstrap initial competence, and Reinforcement Alignment, which refines the model's responses using full reward-driven optimization. Key components include Curriculum Progress Perception (CPP), which adjusts reward thresholds to match the model's evolving competence, and Curriculum Difficulty Perception (CDP), which prioritizes difficult samples to enhance training efficiency and prevent overfitting.

tal limitation of existing GRPO-based methods: their failure to effectively sample correct responses, leading to inadequate positive reward signals for meaningful learning. To address this, CPP is designed to gradually reduce the difficulty of sampling correct responses while refining target-domain knowledge acquisition. CPP dynamically schedules reward thresholds to match the model's evolving competence—applying relatively low thresholds in the early stage to capture weak signals, and progressively increasing them to enforce stricter performance criteria as the model matures. CDP further enhances training efficiency by prioritizing difficult samples (with greater learning value) and down-weighting simpler ones, thereby preventing overfitting and encouraging high-value learning. Through this progressive and curriculum-driven approach, CRPA effectively coordinates deep knowledge integration with the preservation of existing capabilities, yielding a robust and adaptive alignment strategy.

## 4.2 RL BACKBONE: GRPO FOR NON-DEEP-THINKING MODELS

We adapt the GRPO framework for VLMs that perform short-answer generation tasks (e.g., VQA), and denote this variant as GRPO for Non-Deep-Thinking (GRPON). A central component of GRPON is a rule-based reward function $R(o, y)$, which evaluates a generated output $o$ against a ground-truth answer $y$. The reward integrates three complementary criteria:

**Semantic Similarity** ($S_s(o, y)$): Measured as the cosine similarity between Sentence-BERT (Reimers & Gurevych, 2019) embeddings of $o$ and $y$.

**Factual Consistency** ($S_f(o, y)$): Assessed using a Natural Language Inference (NLI) model (Conneau et al., 2019), which checks bidirectional entailment and contradiction between $o$ and $y$.

**Entity Alignment** ($S_e(o, y)$): Computed as the F1-score over entities extracted by SpaCy (Honnibal et al., 2020) from $o$ and $y$.

The overall similarity score is defined as a weighted combination of these components:

$$S(o, y) = \alpha \cdot S_s(o, y) + (1 - \alpha) \cdot [\beta \cdot S_f(o, y) + (1 - \beta) \cdot S_e(o, y)], \tag{3}$$

where $\alpha, \beta \in [0, 1]$ are tunable hyperparameters that balance the contribution of each factor. This similarity score is then converted into a binary reward using a threshold $\delta$:

$$R(o, y; \delta) = \begin{cases} +1 & \text{if } S(o, y) > \delta, \\ -1 & \text{otherwise.} \end{cases} \tag{4}$$

Within our curriculum-driven framework, the threshold $\delta$ is scheduled dynamically across training steps: it begins at a relatively low value to enforce knowledge acquisition during the early stage, and is gradually increased to encourage precision and correctness as training progresses.

### 4.3 CURRICULUM PROGRESS PERCEPTION

The Curriculum Progress Perception (CPP) module is designed to bootstrap learnable signals during the early stages of adaptation and to gradually transition toward free-form generation. It operates through two key mechanisms: (1) answer-prefix injection and (2) step-level threshold scheduling, with the overall goal of reducing the difficulty of sampling the correct answer and shaping an optimizable output distribution when the model initially lacks domain-specific knowledge.

**Answer-Prefix Injection.** For a training sample $\mathbf{x}_i = \{\text{image}_i, \text{prompt}_i\}$ with ground-truth answer $y_i = (y_{i,1}, ..., y_{i,|y_i|})$, let $s$ denote the current training step. CPP injects a prefix of length $k(s) = max(0, (1 - \frac{s}{S} \times \sigma)) \cdot |y_i|$ into the input context, where $\sigma$ is the sliding token ratio. Each sample is treated as one training step, so $\sigma$ controls the overall number of steps in the Pre-Alignment Phase. The resulting context at step $s$ becomes $C(i, s) = (\text{image}_i, \text{prompt}_i, y_{i, \leq k(s)})$, and the model is required to generate only the suffix $y_{i, > k(s)}$. At $s = 0$, the model predicts only the EOS token, minimizing risk. At $s \geq S/\sigma$, no prefix is provided, reducing the task to standard full-response generation. This injected prefix acts as an explicit supervision anchor that strengthens cross-modal attention between the image and the partial textual answer, while also increasing the density of rewardable samples by reducing invalid completions, particularly crucial during early adaptation.

**Step-Level Threshold Scheduling.** To enhance progressive generation, CPP regulates the reward threshold $\delta$ (Equation 4). In the early stages of training, when the model is exposed to simpler tasks, a lower threshold $\delta_{\min}$ is used, which encourages exploration and allows the model to generate more diverse outputs so as to better acquire domain knowledge. As training progresses and the model gains more domain knowledge, the threshold is gradually increased to $\delta_{\max}$, enforcing stricter alignment with the injected prefix to ensure accurate generation. This progression ensures that the model moves from broad exploration to focused, precise output generation. The threshold schedule follows a linear progression, with $\delta(s)$ formalized as:

$$\delta(s) = \delta_{\min} + (\delta_{\max} - \delta_{\min}) \times min(1, \frac{s}{S} \times \sigma), \tag{5}$$

where $\delta_{\max}, \delta_{\min}$ are predefined clipping parameters. The GRPON objective is applied only to suffix tokens ($t > k(s)$), conditioned on the context $C(s)$ and step-dependent threshold $\delta(s)$.

### 4.4 CURRICULUM DIFFICULTY PERCEPTION

The Curriculum Difficulty Perception (CDP) module dynamically reweights training samples based on their difficulty, as reflected in the model's real-time learning state. By prioritizing challenging samples and down-weighting simpler ones, CDP ensures that training resources are allocated to examples with the greatest learning value, thereby improving efficiency and mitigating overfitting.

For a given query, let $\mathbf{r} = (r_1, \ldots, r_G)$ denote the rewards obtained from $G$ generated responses. The mean reward $\overline{r} = \text{mean}(\mathbf{r})$ serves as a proxy for sample difficulty: higher values indicate that the sample is easy, while lower values indicate greater difficulty. The sample weight $w$ is defined as $w = \frac{1}{\text{offset} + \overline{r}}$. This weight is incorporated into the GRPON objective by scaling the advantage term $\hat{A}_{i,t}$ for each sample, ensuring that the model allocates greater capacity to high-value learning opportunities. We set the offset to 1.5 to scale $w$ into a reasonable range, thereby stabilizing the training process. At training step $s$, the complete CRPA objective is expressed as:

$$\mathcal{J}_{\text{CRPA}}(\theta; s) = \mathbb{E}_{x \sim P(X), \{o_i\}_{i=1}^G \sim \pi_{\theta_{\text{old}}}(O|C(s))}$$

$$\frac{1}{G} \sum_{i=1}^{G} \frac{1}{|o_i| - k(s)} \sum_{t=k(s)+1}^{|o_i|} \left\{ \min\left( \rho_{i,t}(\theta)\hat{A}_{i,t}, \ \text{clip}(\rho_{i,t}(\theta), 1 - \epsilon, 1 + \epsilon)\hat{A}_{i,t} \right) \cdot w \right.$$

$$\left. - \gamma \, \mathbb{D}_{\text{KL}}\left[ \pi_\theta(\cdot \mid C(s)) \, || \, \pi_{\text{ref}}(\cdot \mid C(s)) \right] \right\},$$

$$\tag{6}$$

---

**Algorithm 1** Curriculum-Driven Reinforcement Pre-Alignment (CRPA)

---

**Input:** Pre-trained VLM $\pi_{\theta_{\mathrm{pre}}}$; Target dataset $\mathcal{D}_{\mathrm{target}} = \{(\mathbf{x}_i, y_i)\}_{i=1}^{N}$; Curriculum parameters $(S, \delta_{\max}, \delta_{\min}, \sigma)$; GRPO hyperparameters $(\alpha, \beta, \epsilon, G)$.

**Output:** Domain-adapted model $\pi_\theta$.

1: Initialize $\pi_\theta \leftarrow \pi_{\theta_{\mathrm{pre}}}$, reference model $\pi_{\theta_{\mathrm{ref}}} \leftarrow \pi_{\theta_{\mathrm{pre}}}$, step $s \leftarrow 0$, $S \leftarrow N$
2: Sample minibatch $B \subset \mathcal{D}_{\mathrm{target}}$
3: **for** sample $(\mathbf{x}, y) \in B$ **do**
4:     Compute prefix length $k(s) = max(0, (1 - \frac{s}{S} \times \sigma)) \cdot |y|$
5:     Construct context $C(s) = (\mathbf{x}, y_{\leq k(s)})$
6:     Generate $G$ candidate outputs $\{o_i\}_{i=1}^{G} \sim \pi_\theta(\cdot | C(s))$
7:     **for all** $o_i \in \{o_i\}_{i=1}^{G}$ **do**
8:         Compute reward $R(o_i, y; \delta(s))$ using Equation 4
9:         Compute mean reward $\bar{r} = \mathrm{mean}(\{R(o_i, y; \delta(s))\}_{i=1}^{G})$
10:        Compute difficulty weight $w = \frac{1}{\mathrm{offset} + \bar{r}}$
11:        Compute curriculum threshold $\delta(s) = \delta_{\min} + (\delta_{\max} - \delta_{\min}) \times min(1, \frac{s}{S} \times \sigma)$
12:        Compute CRPA objective $\mathcal{J}_{\mathrm{CRPA}}(\theta; s)$ using Equation 6
13:        Update $\pi_\theta$ by gradient ascent on $\mathcal{J}_{\mathrm{CRPA}}(\theta; s)$
14:     **end for**
15:     s = s+1
16: **end for**
17: **return** $\pi_\theta$

---

where the advantage $\hat{A}_{i,t}$ is computed using the reward $R(o_i, y; \delta(s))$. The overall training procedure is summarized in Algorithm 1.

## 5 EXPERIMENT

We begin this section by outlining our experimental setup (Section 5.1). We then present a comprehensive comparison of CRPA against current state-of-the-art methods (Section 5.2). This is followed by ablation studies that assess the contribution of each core component (Section 5.3) and parameter sensitivity analyses that evaluate the robustness of key hyperparameters (Section 5.4). Additional results on computational efficiency, the effectiveness of cold start in RL, and optimization stability and generalization are provided in Appendix A.

### 5.1 EXPERIMENTAL SETUP

**Benchmark Datasets.** To evaluate CRPA's adaptability and performance, we conduct experiments on three benchmark datasets spanning diverse domains: image captioning, geometric problem-solving, and medical X-ray diagnostics.

- COCO Caption (Chen et al., 2015): A widely used dataset for image captioning, containing 123,287 images. We use the original training and test splits of the dataset.

- Geo170K (Gao et al., 2025): A dataset for geometric problem-solving. It is divided into Phase 1 (non-deep thinking, direct-answer tasks) and Phase 2 (deep thinking, multi-step tasks). We use Phase 1, which contains 60,252 samples, with 57,252 for training and 3,000 for testing.

- OpenI (Demner-Fushman et al., 2012): A chest X-ray diagnostic dataset with 6,423 images and corresponding radiological reports. The task is to generate concise and clinically accurate diagnostic descriptions directly from X-ray images. The training set consists of 5,423 images, while the test set includes 1,000 images.

These datasets enable us to evaluate CRPA's performance across both general domain (e.g., COCO Caption) and specific domain (e.g., Geo170K, OpenI) tasks, providing a comprehensive assessment of its domain-adaptive capabilities.

**Baselines.** We employ Qwen2.5-VL-7B (Bai et al., 2025) as the base model and compare the following adaptation strategies. (1) **BASE**: Direct inference using the pre-trained Qwen2.5-VL-7B model. (2) **SFT-based Methods**: Including Parameter-Efficient Fine-tuning (PEFT) via LoRA (Hu et al., 2021) and Full Fine-tuning (FFT). Furthermore, drawing on relevant methods in incremental learning (Kirkpatrick et al., 2016), we incorporate the Kullback-Leibler (KL) divergence loss into

| Datasets | Methods | Domain-Specific Ability | | | | General-Purpose Ability | | | |
|---|---|---|---|---|---|---|---|---|---|
| | | BLEU-1 | ROUGE-L | CIDEr | SPICE | MMMU | MME | IFEval-P | IFEval-I |
| COCO Caption | BASE | 0.4457 | 0.3672 | 0.2259 | 0.1783 | 0.5122 | 2333.36 | 0.6211 | 0.7038 |
| | PEFT | 0.3722 | 0.3081 | 0.1231 | 0.0862 | **0.6067** | **2448.67** | 0.5416 | 0.6535 |
| | FFT | **0.7581** | **0.5474** | **1.0172** | **0.2385** | 0.4244 | 735.10 | 0.2070 | 0.3405 |
| | CFFT | 0.6518 | 0.4824 | 0.7654 | 0.2034 | 0.4967 | 1934.32 | 0.5324 | 0.6419 |
| | GRPO | 0.2245 | 0.2767 | 0.2699 | 0.1297 | 0.5222 | 2301.53 | 0.6506 | 0.7410 |
| | DAPO | 0.2431 | 0.2798 | 0.2687 | 0.1301 | 0.5111 | 2330.27 | **0.6577** | **0.7423** |
| | GRPON | 0.4437 | 0.3656 | 0.3189 | 0.1790 | 0.5100 | 2315.97 | 0.6299 | 0.7238 |
| | CRPA | 0.7478 | 0.5432 | 0.9814 | 0.2383 | 0.5278 | 2289.18 | 0.6470 | 0.7326 |
| Geo170K | BASE | 0.3859 | 0.3014 | 0.2740 | 0.2901 | 0.5122 | 2333.36 | 0.6211 | 0.7038 |
| | PEFT | 0.0901 | 0.1192 | 0.0009 | - | **0.6067** | **2449.67** | 0.5360 | 0.6535 |
| | FFT | **0.6098** | **0.5526** | **2.3109** | **0.5627** | 0.4667 | 2172.37 | 0.5693 | 0.6451 |
| | CFFT | 0.5719 | 0.5091 | 1.8827 | 0.5079 | 0.5078 | 2199.59 | 0.5888 | 0.6676 |
| | GRPO | 0.3799 | 0.3113 | 0.2661 | 0.2878 | 0.5022 | 2346.08 | 0.6373 | 0.7131 |
| | DAPO | 0.3835 | 0.3189 | 0.2776 | 0.2989 | 0.5111 | 2319.25 | 0.6285 | 0.7110 |
| | GRPON | 0.4086 | 0.3431 | 0.3543 | 0.3350 | 0.5122 | 2320.54 | 0.6414 | 0.7062 |
| | CRPA | 0.5998 | 0.5501 | 2.2821 | 0.5623 | 0.5122 | 2315.37 | **0.6414** | **0.7278** |
| OpenI | BASE | 0.1155 | 0.1299 | 0.0002 | 0.0988 | 0.5122 | 2333.36 | 0.6211 | 0.7038 |
| | PEFT | 0.0786 | 0.0977 | - | 0.0871 | **0.6067** | **2449.67** | 0.5508 | 0.6631 |
| | FFT | **0.3396** | **0.2399** | **0.0903** | **0.1900** | 0.4111 | 1623.35 | 0.5323 | 0.6367 |
| | CFFT | 0.2698 | 0.1889 | 0.0813 | 0.1698 | 0.4711 | 2100.32 | 0.5578 | 0.6719 |
| | GRPO | 0.1179 | 0.1309 | 0.0003 | 0.0994 | 0.5122 | 2356.23 | 0.6248 | 0.7062 |
| | DAPO | 0.1165 | 0.1356 | 0.0003 | 0.0998 | 0.5111 | 2334.65 | 0.6267 | 0.7098 |
| | GRPON | 0.1182 | 0.1311 | 0.0003 | 0.0999 | 0.5044 | 2315.37 | 0.6192 | **0.7110** |
| | CRPA | 0.3342 | 0.2325 | 0.0886 | 0.1814 | 0.5011 | 2321.40 | **0.6285** | 0.7062 |

Table 1: Performance comparison of different recommendation methods in terms of Domain-Specific Ability and General Ability. COCO Caption, Geo170k, and OpenI are used as benchmark datasets. The results across these datasets demonstrate that CRPA achieves domain-specific performance comparable to that of FFT, while also preserving general capabilities. '-' indicates that the metric is either too small or the generated output is too long, causing it to be truncated and thus unable to be calculated. The best results are shown in bold. The second-best results are marked with an underline.

Full Fine-Tuning (FFT), and define this improved approach as Continual Full Fine-Tuning (CFFT). Specifically, this KL divergence loss imposes a constraint that aligns the output distribution of the fine-tuned model with that of the pre-trained model, effectively mitigating the overwriting of general capabilities by domain-specific signals. (3) **RL-based Methods**: Group Relative Policy Optimization (GRPO) (Shao et al., 2024), Decoupled Clip and Dynamic sAmpling Policy Optimization (DAPO) (Yu et al., 2025), GRPO for Non-Deep-Thinking Models (GRPON). Cold start is added to GRPON to evaluate our method comprehensively.

**Evaluation Metrics.** We evaluate CRPA using a combination of task-specific metrics for domain knowledge injection and generalization metrics for preserving general-purpose performance. Domain-specific ability is assessed with **BLEU-1** (Papineni et al., 2002), **CIDEr** (Vedantam et al., 2014), **ROUGE-L** (Lin, 2004), and **SPICE** (Anderson et al., 2016) to measure n-gram overlap, semantic consistency, long-sequence similarity, and structural alignment. General-purpose capabilities are evaluated using **MMMU** (Fu et al., 2023) for cross-domain reasoning, **MME** (Fu et al., 2023) for multimodal understanding, and **IFEval** (Zhou et al., 2023)—including IFEval-Prompt (IFEval-P) and IFEval-Instruct (IFEval-I)—to measure instruction-following fidelity and consistency with human intent.

**Implementation Details.** For model training and inference, we set the hyperparameters $\alpha$ and $\beta$ in Equation 3 to 0.6 and 0.7, respectively. In Equation 5, the threshold parameters are configured with $\delta_{max} = 0.7$, $\delta_{min} = 0.8$, and $\sigma = 16$. For the CRPA objective (Equation 6), the regularization coefficient $\gamma$ is set to 0.01. All experiments are implemented using the EasyR1 RL framework (Zheng et al., 2025b) and LlamaFactory SFT framework (Zheng et al., 2024) for VLMs and conducted on a Linux-based server equipped with NVIDIA GPUs.

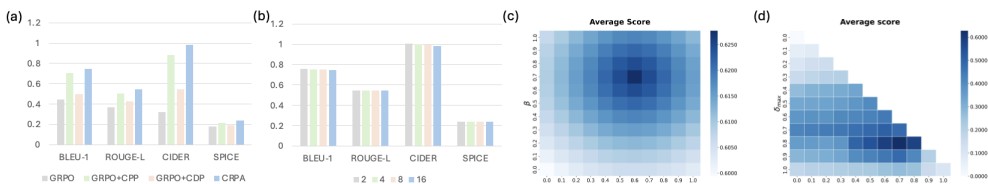

Figure 2: Results of Ablation and Parameter studies. (a) Ablation on COCO Captions demonstrates the contribution of CPP and CDP, with CPP leading to larger gains in domain-specific learning, while CDP enhances specialization by reweighting samples based on difficulty. (b) The impact of the sliding token ratio ($\sigma$) on domain knowledge learning and training efficiency reveals the optimal value of 16. (c) and (d) Parameter study results show that optimal performance is achieved with $\alpha = 0.6, \beta = 0.7, \delta_{min} = 0.7$, and $\delta_{max} = 0.8$, based on aggregated evaluation metrics.

## 5.2 PERFORMANCE COMPARISON

We summarize the main results in Table 1. Overall, CRPA delivers competitive domain-specific performance while better preserving general capabilities. On the COCO Caption dataset, several key observations emerge: (1) FFT delivers the strongest domain-specific performance but causes severe degradation in general capabilities, with declines of -8.78% on MMMU, -1598.26 on MME, -41.41% on IFEval-P, and -36.33% on IFEval-I. (2) PEFT via LoRA mitigates degradation in general metrics such as MMMU and MME, but it significantly reduces instruction-following ability on IFEval (-7.95% on IFEval-P and -5.03% on IFEval-I) and remains less effective at acquiring domain-specific knowledge. (3) GRPON surpasses standard GRPO on domain-specific captioning, yielding gains of +21.92% (BLEU-1), +8.89% (ROUGE-L), +4.9% (CIDEr), and +4.93% (SPICE). These improvements result from avoiding unnecessary deep-thinking iterations and focusing on direct caption generation for non-deep-thinking tasks. (4) CFFT method can mitigate the degradation of generalization ability to a certain extent, but it merely achieves a trade-off and is suboptimal for both domain-specific performance and the preservation of generalization ability. (5) CRPA matches FFT on domain-specific metrics while markedly preserving general-purpose abilities, striking a better balance between specialization and generalization.

On Geo170K and OpenI datasets, we observe consistent trends. CRPA achieves domain-specific performance on par with FFT while substantially outperforming FFT in preserving general multimodal understanding. LoRA preserves some generality but underperforms in domain knowledge acquisition, whereas GRPON improves over GRPO in domain-specific alignment yet falls short of CRPA's balance. These results demonstrate that CRPA offers a practical and effective solution for domain adaptation: it approaches the domain performance of FFT without suffering catastrophic forgetting, and it outperforms alternative post-training methods in balancing domain specialization with general capability preservation.

## 5.3 ABLATION STUDY

To evaluate the contribution of CRPA's key components, we conduct ablations on the COCO Captions dataset (see Figure 2 (a)). Both Curriculum Progress Perception (CPP) and Curriculum Difficulty Perception (CDP) contribute to improved domain-specific learning, with CPP yielding the larger gains. CPP enhances adaptation by injecting partial answer prefixes and progressively shortening them across training steps. This strategy decomposes full response generation into staged subtasks—ranging from predicting only the EOS token to producing complete answers. By this, CPP lowers the early optimization barrier when domain expertise is limited, increases the density of rewardable trajectories, and strengthens cross-modal grounding between images and partially revealed answers. In contrast, CDP improves specialization by dynamically reweighting samples according to difficulty, emphasizing underlearned instances with low similarity or negative rewards and de-emphasizing overlearned easy cases. This mechanism prevents overfitting to narrow domain patterns, encourages broader coverage of underrepresented cases, and preserves flexibility for generalization within the target domain. Together, CPP and CDP constitute a complementary curriculum: CPP optimizes the learning path via progressive guidance and reward densification, whereas CDP

sharpens the learning focus through difficulty-aware prioritization, with CPP playing the leading role in driving efficient domain knowledge injection.

### 5.4 PARAMETER STUDY

**Impact of sliding token ratio.** As shown in Figure 2 (b), a larger $\sigma$ means the model will restore answer tokens at a faster pace. While this accelerates training by enabling the model to complete full response generation in fewer iterations, it undermines domain-specific learning: the rapid recovery leaves insufficient time for the model to thoroughly absorb and align with domain knowledge, resulting in only superficial mastery of key features. In contrast, a smaller ratio prolongs the staged learning process, facilitating deeper domain knowledge acquisition but at the cost of reduced training efficiency. To balance these trade-offs—ensuring sufficient knowledge integration while maintaining practical efficiency—we empirically select $\sigma = 16$ as the optimal setting. This configuration slows token recovery just enough to support incremental domain knowledge learning, while keeping training computationally feasible.

**Influence of weight and threshold parameters:** To assess the effect of weight and threshold parameters, we evaluate performance under varying $\alpha$, $\beta$, $\delta_{\min}$, and $\delta_{\max}$ (Figures 2(c) and (d)). Since individual metrics (BLEU-1, CIDEr, ROUGE-L, SPICE) may emphasize different aspects of generation quality, we aggregate them into a single averaged score for evaluation. For each parameter, we fix one value and vary the other from 0 to 1 with a step size of 0.1. The results indicate that the best performance is achieved at $\alpha = 0.6$ and $\beta = 0.7$, which balance semantic similarity, factual consistency, and entity alignment in the reward function. Similarly, the optimal thresholds for progressive generation occur at $\delta_{\min} = 0.7$ and $\delta_{\max} = 0.8$. These settings are therefore adopted in all subsequent experiments.

## 6 LIMITATION

While CRPA demonstrates strong performance in challenging domain adaptation settings, it offers only marginal improvements over standard supervised fine-tuning when the target domain closely resembles the pretraining distribution, as catastrophic forgetting is minimal in such low-shift settings. Furthermore, CRPA is currently tailored to generative, question-answering style tasks and is not directly applicable to non-generative vision-language tasks (e.g., classification or retrieval), limiting its scope to open-ended multimodal generation. These constraints define the current applicability boundary of our method, and we leave extensions to broader task formats for future work.

## 7 CONCLUSION

We propose Curriculum-Driven Reinforcement Pre-Alignment (CRPA), a novel framework for domain adaptation in VLMs that addresses the challenge of acquiring domain-specific knowledge without forgetting general capabilities. CRPA uses a two-phase process: pre-alignment to stabilize knowledge acquisition and reinforcement alignment to refine behavior. The Curriculum Progress Perception (CPP) and Curriculum Difficulty Perception (CDP) modules guide efficient learning and prevent overfitting. Experiments on domain-specific VQA benchmarks show CRPA achieves strong domain alignment while preserving general multimodal capabilities, outperforming both SFT and RL methods. This work demonstrates the effectiveness of progressive, curriculum-driven strategies for domain adaptation in VLMs.

## 8 REPRODUCIBILITY STATEMENT

We are committed to ensuring the reproducibility of our work. The code and datasets used for all experiments in this paper will be made publicly available upon acceptance. The implementation will be shared on https://github.com/anonymous-debug11/CRPA along with detailed instructions for setting up the environment, training the models, and evaluating the results. We will also include configuration files, hyperparameter settings, and pre-trained model checkpoints to facilitate replication of our experiments. All experiments were conducted on standard hardware, and we will provide detailed

hardware and software specifications. Additionally, we will offer any necessary clarifications and respond promptly to inquiries regarding the methodology and implementation.

To further assist reproducibility, we will provide links to the datasets used in our study, including any pre-processing steps that were applied. We encourage other researchers to validate our findings and contribute to the ongoing development of the model.

## 9    THE USE OF LARGE LANGUAGE MODELS (LLMS)

The Large Language Models were only involved in editing and polishing the language of the paper, improving clarity, consistency, and readability. All scientific content, methods, results, and conclusions are the authors' own work, and the use of the LLMs was limited to editorial tasks, without influencing the scientific integrity or originality of the research.

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

## A APPENDIX

### A.1 COMPUTATION EFFICIENCY

As shown in Figure 3 (a) and (b), the first pre-alignment stage accounts for 28% of the total training time. Compared with GRPO, CRPA is designed with a more sophisticated value function, which consequently increases the computation time per step by 56% under the same batch size. Nevertheless, considering the significant performance gains brought by our method, such computational overhead is well-justified.

### A.2 THE EFFECTIVENESS OF COLD START IN RL

In order to validate the efficacy of cold-start strategies, we conduct controlled experiments by performing cold-start with partial target-domain training data, specifically adopting 0.1 epoch and 1.0 epoch of data for the cold-start phase, respectively. As shown in Table 2, the experimental results reveal two key findings: (1) Even a modest cold-start (0.1 epoch) leads to improved target-domain performance, yet it irreversibly degrades the model's general capabilities—a degradation that cannot be recuperated via subsequent reinforcement learning (RL) fine-tuning. (2) Increasing the cold-start data volume to 1.0 epoch yields marginally superior target-domain performance compared to our

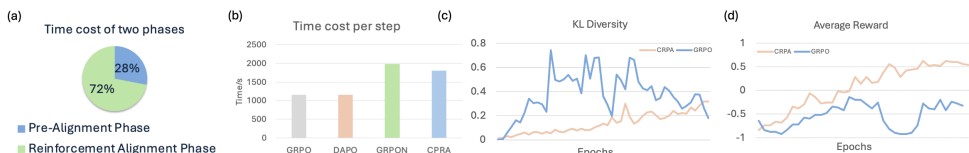

Figure 3: (a) The pre-alignment stage accounts for 28% of the total training time. (b) Under the same batch size, CRPA increases the computation time per step by 56% compared with GRPO. (c) Compared with GRPO, our CRPA reduces policy update variance by about 41% (measured via KL divergence between consecutive policies). (d) Compared with GRPO, our proposed CRPA enables a continuous and stable increment in reward values throughout the entire training process, demonstrating more favorable reward growth characteristics in terms of both sustainability and stability.

| Datasets | Methods | Domain-Specific Ability | | | | General-Purpose Ability | | | |
|---|---|---|---|---|---|---|---|---|---|
| | | BLEU-1 | ROUGE-L | CIDEr | SPICE | MMMU | MME | IFEval-P | IFEval-I |
| COCO Caption | BASE | 0.4457 | 0.3672 | 0.2259 | 0.1783 | 0.5122 | **2333.36** | 0.6211 | 0.7038 |
| | GRPON | 0.4437 | 0.3656 | 0.3189 | 0.1790 | 0.5100 | 2315.97 | 0.6299 | 0.7238 |
| | GRPON+CS(0.1) | 0.6234 | 0.4326 | 0.6776 | 0.1997 | 0.4811 | 1976.37 | 0.5267 | 0.5324 |
| | GRPON+CS(1.0) | **0.7501** | **0.5444** | **0.9921** | **0.2396** | 0.5278 | 901.43 | 0.3012 | 0.4123 |
| | CRPA | 0.7478 | 0.5432 | 0.9814 | 0.2383 | **0.5278** | 2289.18 | **0.6470** | **0.7326** |
| Geo170K | BASE | 0.3859 | 0.3014 | 0.2740 | 0.2901 | 0.5122 | **2333.36** | 0.6211 | 0.7038 |
| | GRPON | 0.4086 | 0.3431 | 0.3543 | 0.3350 | 0.5122 | 2320.54 | 0.6414 | 0.7062 |
| | GRPON+CS(0.1) | 0.5545 | 0.4943 | 1.6893 | 0.4923 | 0.4922 | 2219.54 | 0.6026 | 0.7062 |
| | GRPON+CS(1.0) | **0.6012** | **0.5504** | **2.2896** | **0.5629** | 0.4767 | 2199.34 | 0.5801 | 0.6594 |
| | CRPA | 0.5998 | 0.5501 | 2.2821 | 0.5623 | **0.5122** | 2315.37 | **0.6414** | **0.7278** |
| OpenI | BASE | 0.1155 | 0.1299 | 0.0002 | 0.0988 | **0.5122** | **2333.36** | 0.6211 | 0.7038 |
| | GRPON | 0.1182 | 0.1311 | 0.0003 | 0.0999 | 0.5044 | 2315.37 | 0.6192 | **0.7110** |
| | GRPON+CS(0.1) | 0.2723 | 0.1820 | 0.0823 | 0.1379 | 0.4743 | 1989.34 | 0.5792 | 0.6723 |
| | GRPON+CS(1.0) | **0.3378** | 0.2311 | **0.0896** | **0.1839** | 0.4311 | 1710.69 | 0.5511 | 0.6501 |
| | CRPA | 0.3342 | **0.2325** | 0.0886 | 0.1814 | 0.5011 | 2321.40 | **0.6285** | 0.7062 |

Table 2: Performance comparison of different cold start data volumes in terms of Domain-Specific Ability and General Ability. COCO Caption, Geo170k, and OpenI are used as benchmark datasets. CS(*) means that data of * epoch is used in the cold start. The best results are shown in bold. The second-best results are marked with an underline.

CRPA method, but this gain comes at the expense of substantial generalization degradation (e.g., the MMMU score drops from 0.5122 to 0.4522 on the COCO Caption dataset).

## A.3 OPTIMIZATION STABILITY AND GENERALIZATION

We visualized the Kullback-Leibler (KL) divergence and average reward throughout the training process. As illustrated in Figure 3 (c) and (d), our CRPA method enables stable training: its KL divergence changes in a relatively smooth pattern, and the overall KL divergence is reduced by 41% compared to that of GRPO. Meanwhile, the visualization of the average reward reveals that, benefiting from its targeted design, the average reward of our method steadily rises to a relatively high level—this indicates that our method can effectively learn the knowledge of the target domain. In contrast, GRPO exhibits severe fluctuations in the average reward, and its overall average reward does not increase significantly. All these observations collectively demonstrate the superiority of our proposed method.

