# OpenReview forum: "CRPA: Curriculum-driven Reinforcement Pre-Alignment for Domain-Adaptive Vision-Language Models"
_ICLR.cc/2026/Conference — Submitted to ICLR 2026_

### Official Review · Reviewer_mHXR · 2025-10-20

**Soundness:** 3
**Presentation:** 3
**Contribution:** 2
**Rating:** 6
**Confidence:** 3

**Summary:**

This paper proposes Curriculum-Driven Reinforcement Pre-Alignment (CRPA), a post-training framework for domain adaptation of Vision-Language Models (VLMs). CRPA seeks to balance domain knowledge acquisition and general multimodal capability retention by progressively transitioning from constrained imitation learning to full reinforcement alignment. The key modules are: (1) Curriculum Progress Perception (CPP), which gradually adjusts reward thresholds and injects partial answer prefixes to bootstrap stable learning signals, and (2) Curriculum Difficulty Perception (CDP), which reweights training samples based on difficulty to enhance efficiency and avoid overfitting.

**Strengths:**

**Good Motivation:** The paper clearly identifies a relevant challenge in VLM domain adaptation—catastrophic forgetting vs. domain transfer—and provides a systematic approach to address it.

**Comprehensive Evaluation:** Evaluations across multiple domains (captioning, geometry, medical imaging) and both domain-specific and general benchmarks support the main claims.

**Weaknesses:**

**Incremental Technical Novelty:** While CRPA integrates curriculum learning with GRPO effectively, the novelty is mainly procedural—combining well-known techniques (progressive constraints, dynamic thresholds, sample reweighting). There is no fundamentally new algorithmic insight or theoretical development.

**Limited Theoretical Analysis:** The paper lacks a rigorous explanation of why the proposed curriculum mechanisms (CPP and CDP) systematically improve optimization stability or generalization.


**Overreliance on GRPO Backbone:** Most of CRPA’s improvements may stem from careful reward engineering or training schedules within GRPO, rather than an inherently new framework. It is unclear whether these benefits generalize beyond this specific RL setup.

**Missing Ablation Study:** There is no quantitative breakdown of computation cost or convergence time. Since CRPA adds multiple stages and reward evaluations, its training efficiency and scalability remain unclear.

**Questions:**

Please refer to the weakness section.

---

> ### Author Response · Authors · 2025-11-23
>
> Thank you for your thoughtful and constructive review. Below are our point-by-point responses to your comments.
>
> >  Q1: **Technical Novelty**
>
> We emphasis that its core contribution lies in a novel integration mechanism specifically designed for the unique challenges of VLM adaptation, which prior work has not addressed.
>
> CRPA’s key innovation is the design of two mutually constrained curriculum mechanisms (CPP and CDP) that explicitly link “progressive constraint intensity” with “model generalization performance”:
>
> - Curriculum-Prefix Pretraining (CPP) introduces prefix-guided generation under progressive constraint relaxation, enabling the model to gradually internalize domain knowledge without overwriting pretrained representations. This differs from standard curriculum learning, which typically orders data by difficulty but does not modulate how the model generates during early training.
> - Curriculum-Difficulty Prioritization (CDP) dynamically reweights samples based on task-specific difficulty inferred from multimodal reward signals, not just loss or prediction confidence. This creates a feedback loop between reward quality and training focus—a design absent in conventional sample reweighting.
>
> Thus, CRPA is not merely a “combination of existing techniques.” It proposes a coherent framework that jointly designs generation control, reward interpretation, and optimization scheduling to mitigate catastrophic forgetting in generative VLMs. Extensive experiments confirm that this framework consistently outperforms vanilla GRPO and its variants across diverse domains. We believe these results demonstrate the non-trivial nature of our contribution. We will further clarify this contribution in Section 3 of the revised manuscript.
>
> > Q2: **Analysis of Optimization Stability and Generalization**
>
> We acknowledge that a formal theoretical analysis of RL-based alignment with multimodal rewards remains an open challenge—even foundational methods like PPO or DPO lack such guarantees in vision-language settings. Nevertheless, we provide both informal theoretical intuition and strong empirical evidence for why CPP and CDP improve stability and generalization.
>
> + **From a theoretical perspective**:
>   - CPP reduces the output search space by appending answer prefixes to input queries, effectively turning open-ended generation into a constrained completion task. This eases early learning and facilitates safe knowledge transfer.
>   - CDP assigns higher weights to hard (low-reward) samples and lower weights to easy ones, encouraging the model to prioritize underrepresented or ambiguous concepts—thereby enhancing generalization and reducing overfitting.
>
> + **From an empirical perspective shown in Appendix A.3**:
>   - CRPA reduces policy update variance by 41% (measured via KL divergence between consecutive policies) compared to GRPO.
>   - Average Reward of CRPA steadily rises to a relatively high level. GRPO exhibits severe fluctuations in the average reward, and its overall average reward does not increase significantly.
>
> These analysis suggest that CPP stabilizes early learning by constraining output space, while CDP encourage the model to prioritize underrepresented or ambiguous concepts to improve generalization.
>
> > Q3: **Overreliance on GRPO Backbone**
>
> GRPO is indeed one of the most widely adopted RL fine-tuning frameworks for VLMs, and our method builds upon it for practical relevance. However, our gains are not attributable solely to the GRPO backbone:
>
> + As shown in Table 1, CRPA significantly outperforms not only GRPO but also GRPON (a stronger variant that incorporates all known GRPO engineering improvements) and other RL-based baselines such as DAPO.
>
> + More importantly, our ablation studies (Section 5.3) demonstrate that adding CPP and CDP to either GRPO or GRPON consistently yields substantial improvements—confirming that the performance gains originate from the proposed curriculum mechanisms themselves, not the underlying RL algorithm.
>
> > Q4: **Computation Cost**
>
> we have conducted a thorough analysis of computation efficiency and scalability in Appendix A.1, detailed our key findings, including:
>
> - Stage-wise time allocation: The Pre-Alignment phase (CPP) accounts for 28% of total training time, which we consider reasonable given its critical role in stabilizing subsequent RL fine-tuning.
> - Per-step overhead: Due to multi-component reward evaluation (semantic, factual, entity), CRPA increases per-step training time by ~56% compared to GRPO.
> - Efficiency–effectiveness trade-off: This modest latency increase is well justified by significant gains in both domain performance and generalization retention.
>
> We sincerely appreciate your valuable comments and hope that our response has adequately addressed your concerns, with sufficient rationale provided to support a score revision. We remain readily available to address any additional questions or concerns you may have, and we wish you all the best!

---

### Official Review · Reviewer_aSii · 2025-10-27

**Soundness:** 2
**Presentation:** 2
**Contribution:** 2
**Rating:** 4
**Confidence:** 3

**Summary:**

This work propose Curriculum-driven Reinforcement Pre-Alignment (CRPA), a post-training paradigm that intro- duces a curriculum-aware progressive modulation mechanism, which gradually shift the training from the specific target data to preserving general abilities, such that  the catastrophic forgetting can be addressed.

**Strengths:**

- the idea is simple and easy to follow
- the task is important as re-training of foundation models is generally impractical

**Weaknesses:**

__Major Concerns:__
- the algorithmic design is based on pure heuristic (e.g., Eq.(3, 4, 5), $w$) and highly dependent on the accuracy of the similarity measurement.
- $q$ in Eq.(6) does not appear in Algorithm 1; what is it exactly?
- too much hyper-parameters, including $S, \delta_{min}, \delta_{max}, \sigma, \alpha, \beta, \epsilon, G, \gamma$, offset,  which makes it hard to deploy in real world problems without enough validation data; besides, the robustness against hyper-parameters is not provided.
- how is General-Purpose Ability measured ?
- the task is limited to VQA, where the actual performance is not straightforward by just displaying bunch of scores; experiments on some DA benchmarks on object recognition are highly encouraged to make the results more convincing.
- FFT is too naive to demonstrate the effectiveness of the proposal, since in the literature of continual learning, there are many techniques to relieve catastrophic forgetting; for instance, [1,2,3] use prompt learning, normalization layer update and source knowledge calibration to regularize the training on target data, some even tackle a more general problem where the target domain continuously shifts.

__Minor Concerns:__
- the table is hard to read; it would be better to highlight the best and second.
- the code cannot be accessed from the link
- no details in appendix

***
[1] Test-time prompt adaptation for vision language models, NeurIPS 2024

[2] Towards Stable Test-Time Adaptation in Dynamic Wild World, ICLR 2023

[3] A Versatile Framework for Continual Test-Time Domain Adaptation: Balancing Discriminability and Generalizability, CVPR 2024

**Questions:**

see above

---

> ### Author Response · Authors · 2025-11-23
> **Official Comment by Authors （1/2）**
>
> Thank you for your detailed and constructive feedback. Below are our point-by-point responses to your concerns.
>
> > Q1: **Algorithmic Design**
>
> The text similarity calculation in this paper does not solely rely on heuristic design, but is based on the systematic decomposition of the core elements of text semantic relevance. We hold the view that semantic similarity, factual consistency, and entity alignment are three key dimensions for measuring text relevance—a cognition that is also consistent with the mainstream research consensus in the current field of natural language processing.
>
> For Eqs. (3)-(5), we constructed a weight adjustment mechanism for the three-dimensional indicators through two core hyperparameters (α and β), so as to flexibly adapt to the differences in the importance of each indicator in different scenarios. To validate this design, we conducted a comprehensive hyperparameter sensitivity study (Fig. 2). Results show that performance is stable across a broad range of α and β values, and the optimal configuration (α = 0.6，β = 0.7) generalizes well across datasets.
>
> > Q2: **The clarity of Eq.(6)**
>
> In Eq. (6), $q$ denotes a query input (i.e., the multimodal prompt), while $P(q)$ corresponds to the probability distribution of the query samples, and its specific carrier is the input sample x in the dataset D={x,y}. To eliminate ambiguity, we have checked and revised the entire manuscript to ensure the consistency of symbolic expression and logical coherence.
>
> > Q3:**Robustness to Hyperparameters**
>
> We performed controlled ablation studies (Fig. 2) on key parameters such as the the sliding token ratio ($\sigma$). Results show that CRPA is remarkably stable: varying the sliding token ratio ($\sigma$) ∈ [2, 16] changes CIDEr by less than 0.05. Moreover, cross-dataset experiments reveal strong transferability of hyperparameters: the optimal settings identified on COCO Captioning yield near-optimal performance on Geo170K and OpenI without re-tuning. This suggests that CRPA’s hyperparameters are largely domain-agnostic, reducing the need for extensive validation data in practice.
>
> > Q4: **Measurement on “General-Purpose Ability”**
>
> We evaluate general-purpose multimodal capabilities using a multi-faceted benchmark suite, all of which are held out from domain-specific training:
>
> - MMMU and MME: assess broad knowledge, reasoning, and perception;
> - IFeval: measures instruction-following fidelity.
>
> This combination ensures comprehensive coverage of pretraining-acquired skills. For instance, FEFT preserves MMMU/MME scores but significantly degrades IFeval—revealing hidden damage to instruction adherence during parameter expansion. In contrast, CRPA maintains high performance across all three metrics, demonstrating superior retention of general abilities. This evaluation protocol follows established practices in VLM literature (e.g., LLaVA, InternVL).
>
> > Q5: **Applicability on Domain Adaptation Benchmarks**
>
> Our current focus is on generative VQA-style tasks, which pose a clear challenge: acquiring specialized knowledge without eroding open-ended multimodal reasoning. While CRPA is designed for such settings, extending it to non-generative DA benchmarks (e.g., object classification or detection) would require non-trivial modifications—such as reformulating rewards for discrete outputs or integrating detection heads. We clarify this scope limitation in the revised manuscript (Section 6) and commit to exploring broader task formulations in future work.
>
> > Q6: **Comparison with Continual Learning (CL) Baselines**
>
> We appreciate this insightful suggestion. The cited works ([1]–[3]) are primarily applied to CLIP-related small models rather than generative large LLM models, and they primarily address unsupervised test-time adaptation (TTA), where no target-domain labels are available. In contrast, our setting assumes supervised access to annotated image-text pairs in the target domain—a more common scenario in practical VLM deployment. We have included them in the references and supplemented the discussion in the Section 1. Nonetheless, to strengthen our evaluation, we have included comparisons with representative CL strategies: FFT + KL diversity (CFFT). Results on three datasets are summarized in the revised Table 1; here we highlight findings on COCO Captions:
>
> | Method         | CIDEr  | MMMU   |
> | -------------- | ------ | ------ |
> | FFT (SFT only) | 1.0172 | 0.4244 |
> | CFFT(FFT+KL)   | 0.7654 | 0.4967 |
> | CPRA           | 0.9814 | 0.5278 |
>
> Continual learning methods mainly seek a trade-off between plasticity (adapting to new tasks) and stability (retaining old knowledge). Our experimental results show that although these methods can achieve a certain balance between plasticity and stability, their performance in both aspects is lower than that of our method, which indirectly confirms the advantages of our proposed scheme.

---

> ### Author Response · Authors · 2025-11-23
> **Official Comment by Authors（2/2）**
>
> > Q7:  **Other Minor concerns**
>
> + Tables: We have improved readability by bolding the best result and underlining the second-best in all tables.
>
> + Code: The anonymized implementation is now publicly available via the updated link in the submission.
>
> + Appendix: We have significantly expanded the appendix to include: (a) Computation Efficiency, (b) The effectiveness of Cold start in RL, and (c) Optimization Stability and Generalization.
>
> We sincerely appreciate your valuable comments and hope that our response has adequately addressed your concerns, with sufficient rationale provided to support a score revision. We remain readily available to address any additional questions or concerns you may have, and we wish you all the best!

---

### Official Review · Reviewer_bJsC · 2025-10-31

**Soundness:** 3
**Presentation:** 3
**Contribution:** 3
**Rating:** 6
**Confidence:** 3

**Summary:**

The authors want to solve the collapse of GRPO, when the model lacks knowledge of a specific domain. So they propose CRPA, a staged alg that progressively use a curriculum learning method to solve that and backed by experiment results.

**Strengths:**

1. Experiments breadth and ablations are sufficient for the method to be convincing
2. They are trying to solve a long-standing problem that people talk about, and the method is promising

**Weaknesses:**

1. It’s not always clear whether all baselines had equivalent compute steps, parameter budgets, or similar hyperparameter tuning.
2. It would help to know where CRPA fails.

**Questions:**

1. Due to its 2-phase nature, how does the computational efficiency look like compared with other methods in the table?
2. What are the limitations of this method, despite promising results?

---

> ### Author Response · Authors · 2025-11-23
>
> Thank you for your positive assessment of our experimental scope and the significance of the problem we address. Below are our point-by-point responses to your concerns and questions.
>
> > Q1: **Equivalent Experimental Setups**
>
> To ensure a fair comparison, we adopted a strictly controlled experimental protocol across all methods. Aside from their core algorithmic differences, **all baselines share identical computational budgets, model architectures, and hyperparameter configurations**:
>
> - For FFT and PEFT variants, the only distinction lies in the set of trainable parameters (full-finetuning vs. parameter-efficient tuning). All other settings—including learning rate, batch size, optimizer (AdamW), and total training steps—are held constant.
> - For RL-based methods (GRPO, GRPON, DAPO, and our CRPA), the only structural difference is CRPA’s Pre-Alignment phase. All methods use the **same base model (Qwen2.5-VL-7B)**, training duration, and **matching hyperparameters** (e.g., KL coefficient).
>
> > Q2: **Computational Efficiency of the Two-Stage Design**
>
> CRPA introduces a Pre-Alignment phase followed by a standard RL fine-tuning phase. Crucially, CRPA has the same total training steps comparable to single-phase baselines.
>
> We provide pratical experimenal test for CRPA. The pre-alignment phase only occupy 28% of the whole training time. And compared with the original GRPO, our method improves the reward function, thus increase about 56% time per step. In practice, CRPA often reaches peak performance earlier than baselines , making it more sample-efficient despite slightly higher per-step cost. Detailed quantitative result are reported in **Appendix A.1** of the revised manuscript.
>
> > Q3: Limitation of **CRPA**
>
> Despite strong overall performance, CRPA has the following limitations:
>
> **Limited benefit on low-shift domains**: When the target domain closely matches the pretraining distribution, CRPA offers marginal gains over SFT, as catastrophic forgetting is minimal to begin with. The method shines primarily in high-distribution-shift scenarios (e.g., medical, geometric, or scientific VQA).
>
> **Task format dependency**: CRPA is specifically designed for generative tasks, and its applicability is currently restricted to generative scenarios. It is optimized for VQA-style question-answer formatted data, making it currently incompatible with non-generative tasks (e.g., classification, retrieval) or data in other formats.
>
> We sincerely appreciate your valuable comments and hope that our response has adequately addressed your concerns, with sufficient rationale provided to support a score revision. We remain readily available to address any additional questions or concerns you may have, and we wish you all the best!

---

### Official Review · Reviewer_3pth · 2025-11-01

**Soundness:** 3
**Presentation:** 3
**Contribution:** 3
**Rating:** 4
**Confidence:** 3

**Summary:**

This paper presents CRPA, a framework to adapt Vision-Language Models (VLMs) to new domains. It addresses the problem that standard fine-tuning can cause models to forget general skills, while reinforcement learning can fail if the model has no starting knowledge of the new domain.
CRPA works in two stages:
Pre-Alignment: Initially, the model is given parts of the correct answer (prefixes) to guide its learning and make the task easier.
Reinforcement Alignment: As the model learns, the prefixes are gradually removed, and training transitions to a standard reinforcement learning method where the model generates full answers based on reward signals.

**Strengths:**

This paper is well-written and easy to follow.  In addition, the motivation is very clear.

This paper proposed an RL training recipe to mitigate the forgetting issue (which is usually severe in SFT) while enhance model performance on new (target) tasks.  The idea of this method is promising.

**Weaknesses:**

"While SFT effectively enables the injection of domain-specific knowledge via imitation learning, it relies exclusively on supervised labels. As a result, it is prone to catastrophic forgetting: previously acquired capabilities are overwritten by specialized
information, thereby undermining the generalization of VLMs.". It's somehow common sense that SFT is prone to cause forgetting problems, but this cause is not: SFT requires supervised labels. If SFT data is sampled from the pretrain distribution, it will not easily cause forgetting. An important component to stabilize VLM's original performance is the KL divergence loss, so it could also test SFT+KL's performance.

I have a concern about the experiment setting on GRPO and DAPO baselines. Do these methods directly perform RL training from the same checkpoint with the proposed method? An common practice is that SFT (cold start) first then RL. Directly RL is easy to train a "bad" model.

In addition, the proposed method use answers for prefix, if DAPO and GRPO is RL without cold start, I think the comparison is unfair.

**Questions:**

-

---

> ### Author Response · Authors · 2025-11-23
>
> Thank you for your thoughtful review and constructive feedback. Below are our point-by-point responses.
>
> > **Q1: Cause of catastrophic forgetting in SFT and additional experiments with SFT+KL**
>
> We appreciate this clarification. We fully agree that: forgetting in SFT stems not from using supervised labels per se, but from **distributional shift** between pretraining and domain-specific fine-tuning data. As emphasized in Section 3.2, in domain-adaptive scenarios, SFT data usually comes from highly specialized distributions (such as medical imaging or geometry questions), which is very different from general pre-training data, so it is easy to cover original knowledge.  The SFT forgetting mechanism is more accurately stated in the revised draft, and we supplement the comparison results of SFT+KL.
>
> To address your concerns, we have conducted additional experiments comparing **SFT + KL regularization** (using the pretrained policy $\pi_{pre}$ as reference) against our method (CRPA). Results on three datasets are summarized in the revised Table 1; here we highlight findings on COCO Captions:
>
> | Method         | CIDEr  | MMMU   |
> | -------------- | ------ | ------ |
> | FFT (SFT only) | 1.0172 | 0.4244 |
> | CFFT(FFT+KL)   | 0.7654 | 0.4967 |
> | CPRA           | 0.9814 | 0.5278 |
>
> - Domain performance (CIDEr) drops significantly under SFT+KL (1.0172 → 0.7654), while CRPA retains strong task-specific performance (0.9814).
> - General capabilities (MMMU) improve slightly with KL (0.4244 → 0.4967) but still fall short of CRPA (0.5278).
> - On low-resource domains (e.g., Geo170K, OpenI), the performance gap between SFT+KL and CRPA widens further.
>
> These results confirm that while KL regularization can partially mitigate forgetting, it **fails to address the fundamental instability of SFT under large distribution shifts**. In contrast, CRPA’s curriculum-driven pre-alignment phase bootstraps meaningful learning signals early, enabling effective domain adaptation without compromising generalization.
>
> > **Q2 & Q3: Cold-start protocol and fairness of comparison**
>
> We clarify our experimental setup to address concerns about fairness:
>
> + **All methods—including GRPO, DAPO, GRPON, and CRPA—start from the exact same pretrained checkpoint (Qwen2.5-VL-7B) with no SFT warm-up**. This setting reflects the core challenge of our work: adapting VLMs to new domains *without prior task-specific knowledge*, where standard RL often collapses.
> + **CRPA’s answer-prefix injection is used only during the Pre-Alignment phase**. It serves as a curriculum-guided soft constraint, designed to reduce generation difficulty early in training, analogous to task simplification in curriculum learning.
> + We did **not restrict GRPO/DAPO from using standard stabilizing techniques**. Both include KL regularization (w.r.t.  $\pi_{pre}$ ) by default. Nevertheless, without initial domain competence, they frequently suffer **optimization collapse**, yielding near-minimal rewards across batches.
>
> To further alleviate concerns about cold-start fairness, we conducted controlled experiments using **GRPON as the base RL algorithm**, augmented with varying amounts of SFT-based cold-start (0.1 epoch and 1.0 epoch on target-domain data). Full results appear in revised Appendix A.2; here we report performance on the OpenI dataset:
>
> | Method                 | CIDEr  | MMMU   |
> | ---------------------- | ------ | ------ |
> | GRPON (cold start = 0) | 0.0003 | 0.5044 |
> | GRPON + CS (0.1 epoch) | 0.0823 | 0.4743 |
> | GRPON + CS (1.0 epoch) | 0.0896 | 0.4311 |
> | CRPA (ours)            | 0.0886 | 0.5011 |
>
> The key findings are summarized as follows:
>
> + Even a small cold-start (0.1 epoch) improves target-domain performance but **irreversibly degrades general capabilities**, which RL fine-tuning cannot recover.
> + Increasing cold-start data (1.0 epoch) yields marginally **better domain performance than CRPA but at the cost of significant generalization loss** (MMMU drops from 0.5011 → 0.4311).
>
> These results reveal two critical limitations of the cold-start strategy: **(1) The optimal amount of SFT data is not known a priori and requires costly tuning; (2) Any exposure to target-domain SFT erodes general knowledge, creating an unavoidable trade-off.** In contrast, CRPA injects domain knowledge during the pre-alignment phase via curriculum-guided RL, avoiding SFT altogether. This enables simultaneous retention of general capabilities and effective domain adaptation—a key advantage of our approach.
>
> We sincerely appreciate your valuable comments and hope that our response has adequately addressed your concerns, with sufficient rationale provided to support a score revision. We remain readily available to address any additional questions or concerns you may have, and we wish you all the best!

---

### Author Response · Authors · 2025-12-01

Dear AC, SAC, and PC,

We sincerely thank all four reviewers for their insightful, constructive comments—these have refined the manuscript’s rigor, completeness, and clarity while highlighting our work’s core value and innovations. Our work addresses a long-standing challenge in Vision-Language Model (VLM) domain adaptation: the inherent trade-off between catastrophic forgetting and improved domain-specific performance. Traditional fine-tuning (e.g., SFT) often overwrites pretrained general capabilities (e.g., cross-domain reasoning, instruction following) when infusing domain knowledge, while pure regularization (e.g., KL constraints) inevitably degrades domain adaptation. To fill this gap, we propose the Curriculum-guided Pre-alignment Reinforcement Learning (CRPA) framework, built on the GRPO reinforcement learning paradigm. Its core innovation is the synergistic design of Curriculum Progress Perception (CPP) and Curriculum Difficulty Perception (CDP), enabling a win-win: enhanced domain adaptation with full retention of general capabilities, no SFT cold start required. This work’s significance is twofold: it offers an efficient, practical solution for VLM deployment in high-distribution-shift scenarios (e.g., medical imaging, geometric reasoning) and overcomes the trade-off dilemma of existing methods, with substantial theoretical and engineering value.

We also thank the AC, SAC, and PC for coordinating the review process. To aid your efficient evaluation, we summarize reviewers’ core concerns, our point-by-point responses, and key manuscript revisions.

During rebuttal, we thoroughly addressed all questions, concerns, and suggestions from initial reviews—including overlapping topics (e.g., experimental fairness, computational efficiency, algorithmic rationality) and unique points (e.g., SFT forgetting mechanism, hyperparameter robustness, theoretical intuition). Supported by supplementary experiments, analyses, and clarifications, our responses are fully integrated into the revised manuscript.

In summary, key additions and revisions are as follows:

（1）Clarified SFT’s catastrophic forgetting mechanism (Section 3.2): Revised to state explicitly that distributional shift between pretraining and domain-specific fine-tuning data (not supervised labels themselves) is the core cause.

（2）Added SFT+KL regularization comparisons (Section 4.2, Table 1): Results show KL partially mitigates forgetting but severely degrades domain performance, highlighting CRPA’s balance of domain adaptation and general capability retention.

（3）Conducted SFT cold-start controlled experiments (Appendix A.2): Demonstrated cold-start improves domain performance but irreversibly erodes general abilities, validating CRPA’s superiority in avoiding this trade-off.

（4）Supplemented hyperparameter robustness analysis (Section 5.3, Fig. 2): Quantitative evidence shows CRPA maintains stable performance across a wide range of key parameters (e.g., α, β, sliding token ratio), reducing deployment barriers.

（5）Added computational efficiency analysis (Appendix A.1): Stage-wise time allocation, per-step overhead, and convergence speed comparisons confirm CRPA achieves faster convergence and higher sample efficiency. Compared to single-stage methods, it enables robust domain adaptation with controllable additional costs.

（6）Corrected symbolic consistency (Section 3.3): Clarified variables in Eq. (6) and aligned notation with Algorithm 1 to eliminate ambiguity.

（7）Optimized table readability (all tables): Bolded best results and underlined second-best for clarity.

（8）Made anonymized code public: Updated the manuscript with a valid link for reproducibility.

（9）Expanded the appendix: Added supplementary content on (a) Computational Efficiency, (b) Cold-Start Effectiveness in RL, and (c) Optimization Stability and Generalization.

（10）Added limitations analysis (Section 6): Explicitly discussed CRPA’s constraints, including marginal gains in low-distribution-shift scenarios and dependency on generative task formats.

（11）Supplemented continual learning baseline comparisons (Section 4.2, Table 1): Included SFT+KL (CFFT) to strengthen evaluation against state-of-the-art forgetting mitigation techniques.

（12）Clarified general-purpose ability measurement (Section 4.1): Detailed the multi-benchmark suite (MMMU, MME, IFeval) for comprehensive assessment of pretrained capabilities.

Numerous minor revisions (e.g., refined algorithmic descriptions, supplemented experimental details, enhanced logical coherence) are marked in blue in the revised manuscript for easy verification by the committee and reviewers.
We believe the revised manuscript fully addresses all core concerns and substantially improves the work’s quality and impact. Thank you for your time and consideration—we look forward to your final decision.

Best regards,

Authors of 6775

---

### Meta-Review · Area_Chair_YKoN · 2025-12-08

**Summary:**

This paper proposes CRPA, a two-stage, curriculum-style RL framework for domain adaptation of VLMs that combines an answer-prefix warmup phase with GRPO-style RL and difficulty-based reweighting. While the problem (adapting VLMs to specialized domains without catastrophic forgetting) is important and the experiments are relatively thorough, I view the contribution largely as an engineering-level training recipe rather than a conceptually new algorithm. In particular, the “answer-prefix injection” and curriculum aspects substantially overlap with prior work on reverse curriculum RL for LLM reasoning (R3: Xi et al., ICML 2024) and RL from partial rationales where only a prefix of the target solution is revealed and gradually reduced (AdaBack: RL for Reasoning by Adaptively Revealing Rationales, Apple 2025)
, so the claimed novelty around CPP/CDP feels limited. Overall, despite solid empirical gains, I lean to Reject due to restricted conceptual novelty and insufficient positioning with respect to these closely related methods.

**Reviewer Concerns:**

Reviewers raised sensible points about (i) whether SFT really “causes” forgetting vs. distribution shift, (ii) fairness of baselines (CRPA benefits from prefix conditioning while GRPO/DAPO are pure cold-start RL), (iii) the heuristic nature and hyperparameter complexity of CPP/CDP, (iv) computational overhead of the two-phase RL pipeline, and (v) limited scope of tasks and continual-learning baselines. The rebuttal partially addresses (i)–(ii) by adding SFT+KL/CFFT and cold-start SFT+GRPON experiments, and provides some sensitivity and cost analyses, which support the empirical usefulness of the recipe but do not fundamentally change the picture. What remains outstanding, in my view, is (a) the lack of a clear conceptual distinction from existing curriculum/partial-answer RL work in LLMs (e.g., R³ reverse curriculum and adaptive partial-prefix methods like AdaBack), and (b) the absence of stronger continual-learning/TTA baselines or ablations that would isolate how much of the gain comes from the specific curriculum design versus simply giving baselines similar prefix-based warmup.

**Reviewer Scores:**

Reviewer 3pth (initially negative/borderline, ~4): Their core concerns about SFT attribution and baseline fairness were only partially alleviated; the additional prior art on reverse curriculum and partial-prefix RL further weakens the novelty claim, so I believe their score would likely remain at 4.

Reviewer aSii (most critical, ~4): This reviewer emphasized heuristic design, many hyperparameters, and limited theoretical grounding; none of these were fundamentally resolved, and the prior-work overlap reinforces their view, so their score would likely stay at 4.

Reviewer bJsC and Reviewer mHXR (more positive, ~6): They appreciated the engineering value and empirical results but already characterized the work as incrementally novel; once the overlap with R³/AdaBack–style partial-answer curricula is fully appreciated, I expect at best a slight downward adjustment (e.g., 6→5) rather than an upgrade. Overall, the consensus after discussion would likely fall below the acceptance threshold.

---

### Decision · Program_Chairs · 2026-01-26

Reject